# Physician behavior for "invisible" treatment; Korean herbal medicine doctor's treatment covered by auto insurance

Changwoo Lee ⓘ *

Visiting Scholar, Department of Economics, Boston University, Boston, Massachusetts, United States of America

* changwooda@gmail.com

## Abstract

### Background

This study examines healthcare utilization patterns and provider behavior under auto insurance coverage in South Korea. We investigate whether the significant growth in insurance claims for Korean Herbal Medicine (KHM) is driven by clinical necessity or supply-side structural incentives, such as physician-induced demand.

### Methods

We analyze the utilization gap between Conventional Medicine (CM) and Korean Herbal Medicine (KHM) using cross-sectional secondary data from the Korea Health Panel (KHP). The decomposition method developed by Chernozhukov et al. (2013) allows for the decomposition of differences across the entire distribution of medical visits and length of stay (LOS), distinguishing between patient characteristics and provider-side factors.

### Results

The results indicate that the higher utilization of KHM services is primarily attributable to structural factors rather than patient endowments. In the upper deciles of outpatient visits, structural effects accounted for over 100% of the observed difference, suggesting that provider-side incentives are the dominant driver of high-utilization outliers. Similarly, prolonged hospitalization in the KHM sector was largely unexplained by patient characteristics and remained robust after controlling for patient demographics.

### Conclusions

The observed disparities suggest that structural incentives within the KHM sector may influence provider behavior and utilization patterns differently than in the CM

**Data availability statement:** All Korea Health Panel Survey files are available from their database at https://www.khp.re.kr:444/web/data/data.do.

**Funding:** The author(s) received no specific funding for this work.

**Competing interests:** The authors have declared that no competing interests exist.

sector. While these findings are consistent with the theoretical framework of supply-side inducements, further research incorporating direct clinical severity measures is needed to establish definitive causal links. These results show the need for policy interventions targeting reimbursement structures to enhance the efficiency of auto insurance healthcare delivery.

## Introduction

The rapid increase in Korean herbal medicine claims in auto insurance is a significant issue that has emerged in recent years. Total medical use claims in auto insurance were about 1.9 billion dollars in 2019. Among the total claims, Korean herbal medicine claims are approximately 0.8 billion dollars, accounting for 43.2% in 2019 from 23% in 2015 [1]. The disproportionate growth is evident when comparing the increase in total medical use claims, which rose 1.4 times, to the increase in herbal medicine claims, which rose 2.7 times over the five-year period from 2015 to 2019.

The medical costs for a car accident victim are covered by the assailant's compulsory auto insurance, not national health insurance. The auto insurance fee schedule for the providers follows the national health insurance fee schedule. However, the fee schedule for medical services not covered by national health insurance is not well defined in auto insurance. Therefore, suppliers have different incentives for treating patients covered by auto insurance compared to those covered by national health insurance.

However, as Fig 1 indicates, Korean Herbal Medicine (KHM) doctors expand the medical services covered by auto insurance more aggressively than usual medical doctors, even though both types of doctors may engage in opportunistic behavior in these services. This study explores the causes of the rapid increase in Korean herbal medicine use covered by auto insurance for an assaulter in a traffic accident. In particular, we revisit the theoretical background of induced demand and attempt to explain the differences in marginal psychic costs between the two types of doctors, considering the role of opportunism. KHM doctors have a more integrative approach to treatment than conventional medicine (CM) doctors, which may make them less vulnerable to reimbursement cuts. It is challenging for the reviewer to assess the treatment result and refuse reimbursement. Therefore, it may give them more incentives than usual medical doctors to increase their profits through overtreatment [2,3].

We also attempt to empirically identify this difference and demonstrate a structural difference between herbal medicine doctors and conventional medical doctors in Korea. We employ the empirical methods suggested by Chrenozhukov et al. [4] to investigate the hypothesis that KHM doctors are more likely to pursue opportunistic behavior by inducing demand. Among those using medical care covered by the assaulter's auto insurance, those who utilize herbal medicine and those who use conventional medicine are divided. We implement the decomposition of the distribution of medical use between herbal medicine and conventional medicine. We explain the difference in the distribution of medical use among mildly injured patients covered by auto insurance, comparing herbal medicine and conventional medicine, using outpatient data from the 2017 Korea Health Panel (KHP). To analyze differences in

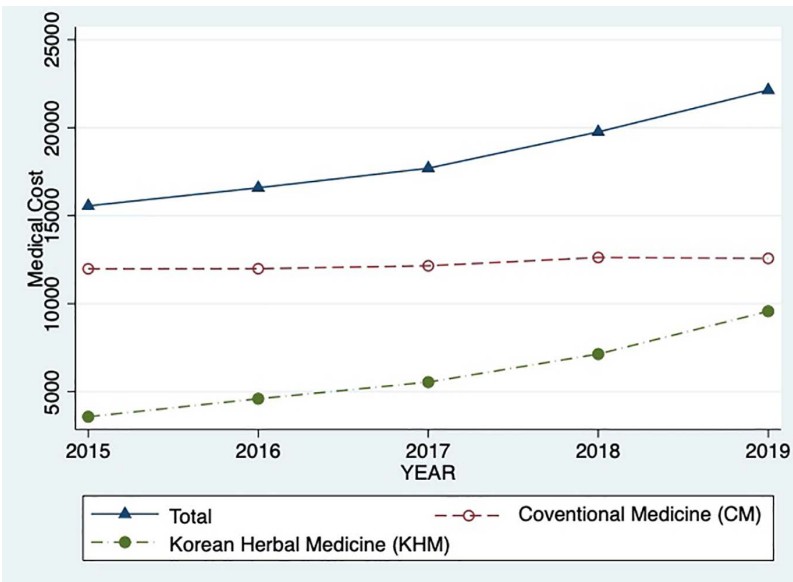

**Fig 1. Trends in Korean herbal medicine claims cost.**

the distribution of medical use covered by auto insurance between herbal and conventional medicine, we use the length of stay (LOS) of inpatients covered by the assailant's auto insurance.

The result shows that the difference between herbal medicine and conventional medical use covered by auto insurance is largest at the upper quantile of the distribution and is mainly attributable to structural differences. This means that the conditional distribution of herbal medicine differs structurally from that of conventional medical use. It might imply that KHM doctors' opportunistic behaviors are more common than those of CM doctors at the upper quantile of medical use distribution.

The rest of the paper is organized as follows. Section 2 presents a theoretical background on why herbal doctors' opportunistic behaviors are strengthened more than those of physicians. Section 3 presents the empirical model, and Section 4 describes the data set. Section 5 presents the result of the decomposition and discusses my findings and the limitations of the study. Finally, Section 6 concludes the paper.

## Theoretical background

### Treatment behavior of physicians

The literature on physicians' opportunistic behavior is mainly on physician-induced demand (PID) [5]. Physicians act as patients' agents to improve patients' health status and pursue their own income and work satisfaction, which are related to the quantity they can set and the prices they charge [6]. According to Santerre and Rexford [5], studies beginning with Newhouse [7], Evans [6], Farley [8], and Fuchs [9] find that physicians sometimes capitalize on their asymmetric information relative to patients by increasing the demand for their services. Therefore, the theory based on PID treats physicians as "rent-seekers" rather than profit-seekers [5]. We refer to McGuire [10] for the theoretical model related to PID. McGuire [10] suggests a physician's utility maximization problem as follows.

$$\text{Max } U = U(Y, I)$$

$$\text{where } Y = N\left(m_1 x_1\left(i_1\right) + m_2 x_2\left(i_2\right)\right), \; I = N(i_1 + i_2)$$

The physician selects the level of inducement, $i$, to maximize her utility which depends on her net income, $Y$ and the total inducement, $I$. $m$ is the margin for each service equal to the difference between the doctor's fee and the service's cost [10].

The utility maximization condition is as follows.

$$m_1 x'_1 = m_2 x'_2 = -\frac{U_I}{U_Y}$$

The marginal (dollar) return to inducement for each service must be equated to the marginal psychic cost (in dollar term) of inducement [10]. Thus the argument of this study holds only if $U_I < 0$, $U_{II} < 0$ and marginal disutility from the inducement of herbal doctors is smaller than marginal disutility of regular doctors, i.e., $U_I^{KHM\ doctor} < U_I^{CM\ doctor}$.

## Medical services in the auto insurance market as in-kind subsidy

Jones [2] presents a theory on the rent-seeking behavior of the producers in the in-kind subsidy market. The in-kind subsidy market could emerge as the altruistic community is generated. According to Jones [2], altruism is difficult to manifest at an individual level. Still, self-interested producers in in-kind subsidy markets exert political pressure for altruistic policies so that altruism can emerge, and, based on this, they establish in-kind subsidy policies. Producers in the in-kind subsidy market participate in the political process of subsidy policymaking, leading to policies that provide subsidies in kind rather than cash, thereby further strengthening their rent-seeking behavior.

Applying Jones [2]'s argument to the current provision of medical services of Korean auto insurance, it can be said that auto insurance, which is compulsory insurance, is a system that provides in-kind medical services to victims of traffic accidents through the perpetrator's auto insurance. Because traffic accident victims are recipients of medical services compensated by the auto insurance of the at-fault driver, these services are a kind of in-kind compensation. A compulsory auto insurance subscription is an expression of an altruistic policy to ensure traffic accident victims receive proper medical services by forcing traffic accident perpetrators to subscribe to auto insurance. It compensates for the health of traffic accident victims in a policy manner. Medical service providers have incentives to pursue rent as producers of in-kind subsidies.

Suppose Jones [2]'s theory is combined with the theory in Section 2.1. In that case, the marginal psychic cost of induced demand for medical providers, directly related to the burden of patients' medical expenses, may be reduced. In a system in which all victims' medical costs are covered by insurance, the moral-psychic costs of opportunistic behavior by medical providers can be alleviated.

## Change in marginal psychic cost of herbal medicine providers

The marginal psychic costs of KHM doctors can be lower than those of CM doctors. The lower marginal psychic costs of KHM doctors may be attributable to transaction costs arising from opportunism. KHM providers have more opportunities in the auto insurance treatment contract than CM doctors due to the lack of information on the safety and effectiveness of herbal medicine, its ingredients, and its origin [11]. Herbal medicine has been developed as a more integrative treatment, based on Chinese meridian theory and using acupuncture, moxibustion, and traditionally recognized herbal medicines, rather than on a scientific assessment of safety and effectiveness. In addition, due to the collective political pressure of KHM doctors, they have been included in health insurance benefits since 2013.

KHM doctors are likely to exploit the legal status of herbal medicine, which allows it to be used without safety and effectiveness evaluation, unlike CM services. Therefore, the incentive for KHM doctors to capitalize on their advantages in the insurance review could appear stronger than that of medical doctors. Since in-kind subsidies for traffic accident victims can reduce marginal psychic costs for KHM doctors, their opportunistic behavior could be more substantial than that of CM doctors.

 

The advantages mentioned above in the insurance review of herbal medicine services motivate the KHM doctor, as an in-kind subsidy producer, to pursue inducing demand more than the CM doctor. Applying Jones's theory [2] to these phenomena, car insurance holders who act as taxpayers or voters in his theory are not interested in finding information on the treatment effects of KHM services because the treatment is for the accident victims, not for them. Therefore, when insurance holders' indifference to the treatment effect of herbal medicine and the advantages of insurance review for herbal medical services are combined, KHM doctors' opportunistic behavior in inducing demand could be achieved more quickly than that of CM doctors. In addition, when auto insurance subscribers are indifferent to political pressure to expand KHM's insurance benefits, KHM doctors can create favorable conditions for their rent-seeking.

## Methods

### Decomposition

This study employs an empirical method proposed by Chrenozhukov et al. [4] to test the hypothesis that herbal doctors may engage in rent-seeking behavior more than medical doctors. The Oaxaca-Blinder (1971) method attempts to decompose the difference between the oriental medicine service and the medical service. The decomposition technique can explain the difference between using herbal medicine and using regular medical services for two reasons. It can be explained by structural differences in medical use arising from differences in the coefficient estimates and from observable characteristics. The structural difference refers to the difference between the conditional probability of using herbal medicine services and that of using regular medical services. This study assumes that the structural difference in medical use between herbal and regular medicine corresponds to a difference in induced demand, i.e., in opportunistic behavior.

While the traditional Oaxaca-Blinder decomposition is limited to explaining differences in the mean of the dependent variable, this study employs the functional decomposition method developed by Chernozhukov et al. [4]. This approach is superior for our analysis because physician-induced demand in the Korean herbal medicine sector often manifests as extreme outliers in the upper deciles of utilization rather than uniform increases across the population. By estimating conditional and unconditional quantile effects, this method allows us to decompose the gap across the entire distribution of medical visits and costs. This enables us to distinguish whether the observed utilization gap is driven by patient characteristics (Endowment Effect) or by provider-side behavioral and incentive structures (Structural Effect) at various levels of treatment intensity.

The decomposition derives from the fact that the marginal distribution of an outcome(Y), in our case, medical use, is equal to the integral of its conditional distribution over the distribution of covariates(X) [12].

$$F_{Y_{KHM}}(y) = \int F_{Y_{KHM}|X_{KHM}}(y|x)dF_{X_{KHM}}(x)$$

Counterfactual distributions can be constructed by integrating the conditional distribution in herbal medicine over the distribution of covariates from regular medicine.

$$F_{Y_{KHM}}^{CM}(y) = \int F_{Y_{KHM}|X_{KHM}}(y|x)dF_{X_{CM}}(x)$$

Implementation of the decomposition requires estimators of the conditional distribution of medical uses and the marginal distributions of its determinants. Distributional regression is used to obtain the estimators. The difference in the distribution of medical use between herbal medicine and regular medicine can then be decomposed as follows:

$$F_{Y_{KHM}}(y) - F_{Y_{CM}}(y) = \left[\hat{F}_{Y_{KHM}}(y) - \hat{F}_{Y_{KHM}}^{CM}(y)\right] + \left[\hat{F}_{Y_{KHM}}^{CM}(y) - \hat{F}_{Y_{CM}}(y)\right] + \hat{\varepsilon}$$

The first term on the right-hand side is the estimated difference in the distribution of medical use attributable to differences in the distributions of its determinants. The second term is the difference due to the structural shifts in the relationship of medical use to its determinants, i.e., the difference in the distribution of medical use conditional on a given distribution of determinants.

We present decompositions of differences in quantiles. These are obtained using the fact that the quantile function is the inverse of the cumulative distribution.

$$Q_{Y_{KHM}}(\tau) - Q_{Y_{CM}}(\tau) = \left[\hat{Q}_{Y_{KHM}}(\tau) - \hat{Q}_{Y_{KHM}}^{CM}(\tau)\right] + \left[\hat{Q}_{Y_{KHM}}^{CM}(\tau) - \hat{Q}_{Y_{CM}}(\tau)\right] + \hat{\nu}$$

## Data

This study is a cross-sectional secondary data analysis utilizing the Korea Health Panel (KHP) data from 2017. KHP, designed to sample from the 2015 Population and Housing Census data to ensure national representativeness, includes data on medical use as well as demographic and socioeconomic information at the individual level. The KHP also contains data on medical use covered by auto insurance, which is the main interest of this study. However, the KHP does not include information on medical expenses incurred for medical use covered by auto insurance, as such detailed items are investigated only when medical use is covered by health insurance. Only data on medical utilization covered by auto insurance are selected and investigated separately, distinguishing between outpatient and inpatient use. For the outpatient covered by auto insurance, 187 observations out of 301,540 are observed, and for the inpatient covered by auto insurance, 95 observations out of 3521 are observed in the 2017 KHP.

## Variables

The number of individual outpatient visits is used as a dependent variable for outpatient utilization. The number of outpatient visits variable is created by identifying individuals who paid medical expenses through auto insurance and summing the number of outpatient cases for which those expenses were paid for each individual. The length of stay (LOS) is a dependent variable for inpatient utilization.

Fig 2 shows the number of outpatient visits (A) and the length of stay (B) for each quantile of people who utilize medical care through auto insurance. The left one shows the number of outpatient visits, and the right one shows the length of hospital stay. There is little difference in the number of outpatient visits between herbal medicine and conventional medical utilization in the low quantiles. However, in the 5th decile, the number of visits to herbal medicine appears higher than that for conventional medical utilization. In addition, the length of hospital stays in herbal medicine appears to be higher than that in conventional medicine in the high quantiles.

Table 1 presents descriptive statistics for the dependent variable and the covariates used in the outpatient visit decomposition. Using auto insurance claims, outpatients visited about eight times on average during 2017. In addition, socioeconomic variables such as male dummy, married dummy, college dummy indicating the respondent have college degree, disability dummy indicating whether the respondent was classified as having a disability by the government; categorical income dummies, age; and labor income level are included as covariates in the decomposition analysis.

Table 2 presents the descriptive statistics for the dependent variable and covariates in the LOS decomposition. According to auto insurance claims, the average length of stay was approximately 11 days in 2017. Socioeconomic factors, as in the outpatient analysis, are covariates in the decomposition analysis.

## Results

Table 3 shows the results of decomposing the difference in quantiles of outpatient use between herbal medicine and regular medicine into the difference attributable to the determinants and the difference attributable to the conditional probability.

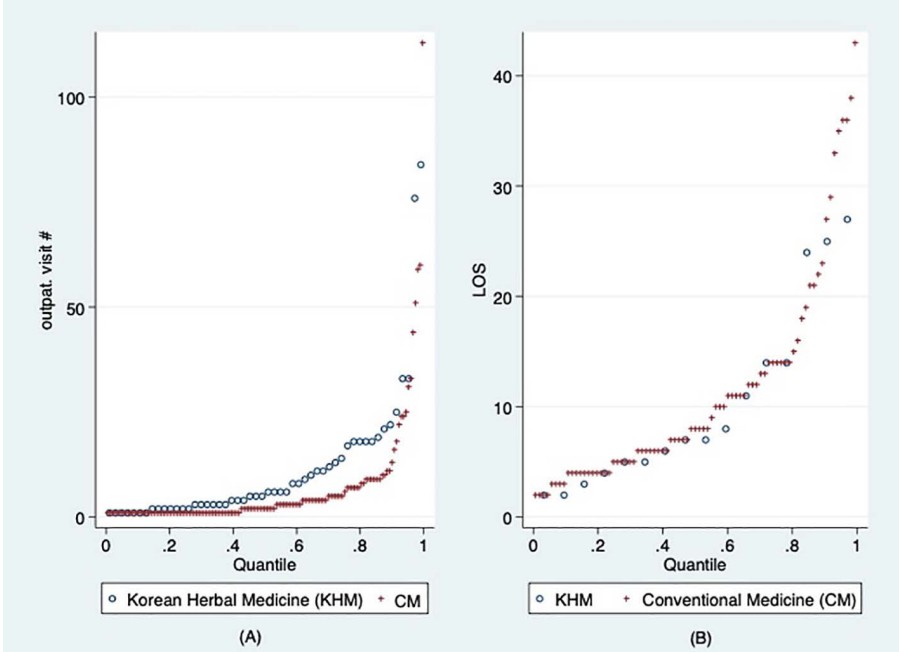

**Fig 2. Distribution of the number of outpatient visits and the length of stay.**

**Table 1. Descriptive statistics of variables for outpatient visit decomposition.**

| Variable(Obs = 187) | Mean | Std. Dev. |
|---|---|---|
| Number of Outpatient Visits | 7.963 | 14.581 |
| Male | .433 | .497 |
| Married | .668 | .472 |
| Age | 48.914 | 18.649 |
| College | .374 | .485 |
| Disability | .096 | .296 |
| Income Quintile 1 | .07 | .255 |
| Income Quintile 2 | .182 | .387 |
| Income Quintile 3 | .251 | .435 |
| Income Quintile 4 | .278 | .449 |
| Income Quintile 5 | .219 | .415 |
| Labor income | 1565.171 | 1969.209 |

Note: For dummy variables, the Mean represents the proportion of the sample.

These results are derived from the distribution regression using a linear probability model. In the overall quantile, the total difference in the number of visits between herbal and regular medicine has a similar pattern, with the difference attributable to structural differences. The direction of the structural difference appears to account for the overall difference across all quantiles.

The decomposition results for outpatient visits indicate that structural factors (conditional distribution) are the primary drivers of the utilization gap, particularly in the higher deciles. Specifically, the ratio of the structural effect to the total

**Table 2. Descriptive statistics of variables for LOS decomposition.**

| Variable(Obs = 95) | Mean | Std. Dev. |
|---|---|---|
| LOS(Length of Stay) | 11.084 | 9.251 |
| Male | .442 | .499 |
| Age | 48.211 | 18.425 |
| Married | .642 | .482 |
| College | .316 | .467 |
| Labor income | 1435.768 | 1705.468 |
| Disability | .105 | .309 |
| Income Quintile 1 | .095 | .294 |
| Income Quintile 2 | .274 | .448 |
| Income Quintile 3 | .179 | .385 |
| Income Quintile 4 | .253 | .437 |
| Income Quintile 5 | .2 | .402 |

**Table 3. Decomposition of differences in quantiles of outpatient visit number b/w Korean herbal medicine and conventional medicine.**

| Quantile | Differences b/w the observable distribution | Difference in covariates | Differences in conditional distribution |
|---|---|---|---|
| 0.1 | 0.349 | −0.182 | 0.531 |
| 0.2 | 1.000 | −0.318 | 1.318 |
| 0.3 | 1.911 | −0.089 | 2.000 |
| 0.4 | 2.802 | −0.127 | 2.929 |
| 0.5 | 4.050 | −0.661 | 4.711 |
| 0.6 | 4.877 | −1.837 | 6.714 |
| 0.7 | 6.697 | −2.263 | 8.960 |
| 0.8 | 9.064 | −0.238 | 9.302 |
| 0.9 | 7.186 | −1.122 | 8.309 |

observed difference shows that structural factors account for 133.8% (8.96 of 6.697 visits), 102.6% (9.302 of 9.064 visits), and 115.6% (8.309 of 7.186 visits) of the differences in the 7th, 8th, and 9th deciles, respectively. These values exceeding 100% signify that the structural gap is wider than the total observed difference, being partially offset by negative endowment effects (patient characteristics) in those quantiles.

The sizable structural difference in the high quantile of outpatient visits may indicate that herbal medicine is more likely to be provided structurally than conventional medical services. These significant structural differences suggest that reimbursement incentives and provider-side factors play a more prominent role in KHM than in CM. This result also shows that the marginal utility of KHM doctors in inducing demand may be greater than that of CM doctors, in line with the theoretical background. While these findings are consistent with the theoretical framework of physician-induced demand, they also show how the uniqueness of KHM's fee review may create structural environments in which utilization patterns deviate significantly from those driven solely by patient need.

If outpatients in traffic accidents generally get mildly injured, the outpatients in the higher quantile of visits are structurally more likely to use herbal medicine than conventional medicine. Therefore, this can explain the rapid increase in the cost of herbal medicine in recent years.

Fig 3, which visualizes Table 1, shows the difference in the quantile function of outpatient visits between herbal and conventional medicine. In the lower quantile, there is little difference in the number of outpatient visits between herbal and conventional medicine. However, as the quantile of outpatient use increases, the difference grows, peaking at the 8th quantile. The structural difference in outpatient visits between herbal medicine and conventional medicine, the main focus of this study, increases at the quantile with high numbers of outpatient visits. The difference in outpatient utilization between herbal and conventional medicine, attributable to the determinants, shows a sharp decrease at the high quantile.

Table 4 shows the results of decomposing the difference in quantiles of the length of hospital stay between herbal medicine and conventional medicine into the difference attributable to the determinants and the difference attributable to the conditional probability. These results are derived from the distribution regression using a linear probability model. In the overall quantile, the difference in LOS between herbal and conventional medicine attributable to determinants follows an opposite pattern, whereas the difference attributable to structural differences is positive. Above the first deciles, the directions of the effects are opposite, and they offset each other's effects. Changes in the relationship of hospital stay to the conditional distribution account for seven-day differences (442%) of a 1.5-day increase in the 7th decile, two days

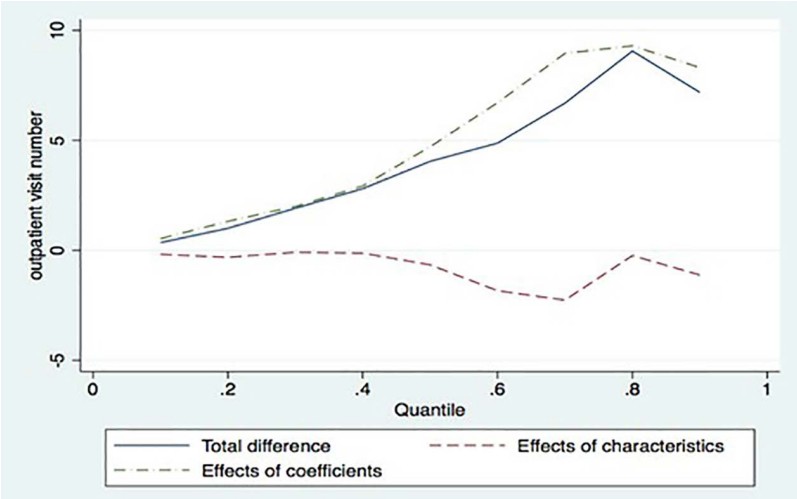

**Fig 3. Decomposition of differences in quantiles of outpatient visits between herbal medicine and conventional medicine.**

**Table 4. Decomposition of differences in quantiles of LOS b/w Korean herbal medicine and conventional medicine.**

| Quantile | Differences b/w the observable distribution | Difference in covariates | Differences in conditional distribution |
|---|---|---|---|
| 0.1 | −1.152 | −1.932 | 0.780 |
| 0.2 | −1.047 | −2.988 | 1.941 |
| 0.3 | −0.500 | −3.500 | 3.000 |
| 0.4 | −0.801 | −4.416 | 3.614 |
| 0.5 | −1.794 | −5.680 | 3.886 |
| 0.6 | −3.312 | −7.898 | 4.586 |
| 0.7 | 1.583 | −5.419 | 7.002 |
| 0.8 | 4.750 | −4.093 | 8.843 |
| 0.9 | −3.011 | −1.534 | −1.477 |

(300%) of a 1-day decrease in the 8th decile, and 8.842 days(186%) of a 4.75-day increase in the 9th decile. The difference in the LOS quantiles between herbal and conventional medicine is positive only at the 7th and 8th quantiles. Across the entire distribution of Length of Stay (LOS), the endowment effect (determinants) consistently shows negative values, suggesting that based on patient characteristics alone, KHM patients would be expected to have shorter stays. However, the structural effect (provider-side factors) is primarily positive and dominant. This indicates that the observed prolonged hospitalization in the KHM sector is not driven by patient health needs but is largely attributable to structural incentives and provider behavior within the herbal medicine framework.

The result shows that the LOS in herbal medicine hospitals is structurally more prolonged than in regular hospitals in the most quantile of LOS. Hospitalization is usually highly severe, so it may show that patients with severe traffic accidents do not use herbal medicine often. It may also indicate that patients who are severely injured in traffic accidents and are hospitalized through herbal medicine will likely stay longer in KHM institutions than in CM hospitals.

This structural difference may suggest that the induced demand for inpatient use in herbal medicine occurs more frequently than in conventional medicine. In other words, the opportunistic behavior of herbal doctors may occur more frequently than that of regular medical doctors, even in inpatient settings. A positive structural difference in the overall quantile of LOS between herbal and conventional medical services may indicate that KHM doctors are structurally more likely to provide medical services than CM doctors.

Fig 4, which visualizes Table 4, shows the difference in the quantile function of the LOS between herbal and conventional medicine. There is little difference in the LOS between herbal and conventional medicine in the lower quantile. As the quantile of LOS increases, the difference jumps to positive but tends to be negative at the last quantile. However, the structural difference in the LOS between herbal and conventional medicine, which is the main interest of this study, is increasing at the most quantile.

## Conclusions

This study investigates the causes of over- and underutilized medical use when a traffic accident victim uses medical care covered by the assailant's car insurance through theoretical background and empirical analysis. Jones's theory [2]

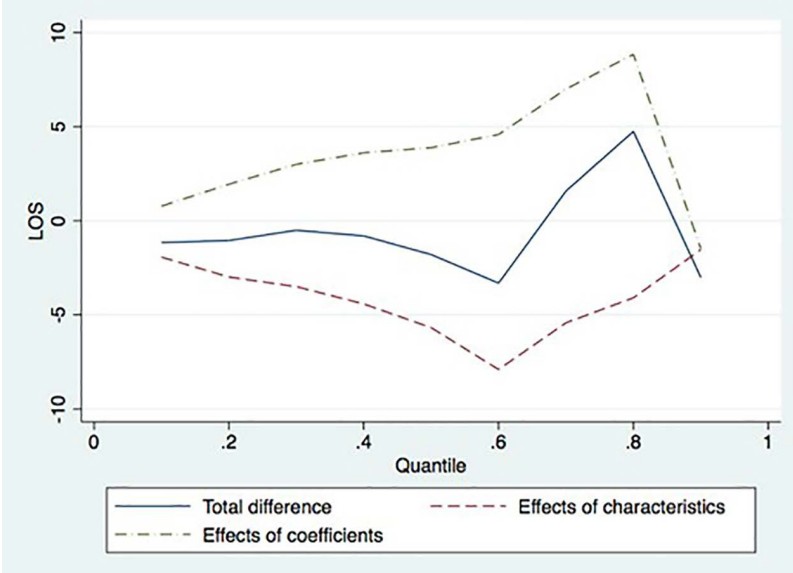

**Fig 4. Decomposition of differences in quantiles of LOS between herbal medicine and regular medicine.**

on strengthening producers' incentives for opportunistic behavior in the in-kind subsidy market confirms that healthcare providers may induce opportunistic behavior toward the traffic accident victim covered by car insurance. Reviewing herbal medical services is complex, and rejection at the review stage is less likely. It is due to uncertainty in treatment fees and recognition criteria, an inadequate system for determining treatment fees, severe information asymmetry, and supervisors' blind spots. The KHM doctors, taking advantage of this, are likely to induce demand for their services, which may be covered by the assailant's auto insurance.

Using the method suggested by Chrenozhukov et al. [4], we examine the hypothesis that KHM doctors' opportunistic behavior is more prevalent than that of CM doctors. The results show a significant structural difference between herbal and conventional medicine in the high quantile of medical utilization. This result indicates that medical services are provided more structurally in herbal medicine than in conventional medicine when a patient seeks the same type of service. However, as this is an observational study using secondary data, these results should be interpreted as identifying associations rather than definitive causal links. Future research incorporating clinical severity measures and direct provider-level data is required to further distinguish between clinical necessity and supply-side inducements.

While the overall KHP is large, the subset of auto-insurance claimants is smaller, which is why the Chernozhukov method is so valuable for extracting meaning from this specific group. Since this study lacks data on the number of inpatient observations covered by auto insurance, the results indicating that KHM doctors may engage in more opportunistic activities through induced demand than medical doctors warrant further investigation in future research. However, the empirical results of this paper suggest that KHM doctors may be influenced by supply-side incentives, as outpatient and inpatient results are consistent.

## Author contributions

**Conceptualization:** Changwoo Lee.

**Data curation:** Changwoo Lee.

**Formal analysis:** Changwoo Lee.

**Investigation:** Changwoo Lee.

**Methodology:** Changwoo Lee.

**Resources:** Changwoo Lee.

**Software:** Changwoo Lee.

**Supervision:** Changwoo Lee.

**Validation:** Changwoo Lee.

**Visualization:** Changwoo Lee.

**Writing – original draft:** Changwoo Lee.

**Writing – review & editing:** Changwoo Lee.

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
