## [Editor Report · Decision Letter 0]

5 Nov 2025

PONE-D-25-53640Physician behavior for “invisible” treatment; Korean herbal medicine doctor's treatment covered by auto insurancePLOS ONE

Dear Dr. Lee,

Thank you for submitting your manuscript to PLOS ONE. After careful consideration, we feel that it has merit but does not fully meet PLOS ONE’s publication criteria as it currently stands. Therefore, we invite you to submit a revised version of the manuscript that addresses the points raised during the review process.

We look forward to receiving your revised manuscript.

Kind regards,

Pasyodun Koralage Buddhika Mahesh

Academic Editor

PLOS ONE

Editor Comments:

The current version of the abstract is less-informative. You are requested to revise it (specially the methods and results sections) before the review process can be further continued.

---

## [Author Response · Author response to Decision Letter 1]

6 Jan 2026

Response to Editor

Comment 1 (Editor):

“The current version of the abstract is less informative. You are requested to revise it (especially the methods and results sections) before the review process can be further continued.”

Response:

We appreciate the editor’s helpful guidance. In response, we have substantially revised the abstract to provide more precise and more detailed information on the study design, data source, empirical method, and main findings. Specifically, we now summarize the dataset used, the decomposition method applied, and the key quantitative results highlighting the structural differences in medical utilization between herbal and conventional medicine.

The updated abstract is as follows:

Background

This study examines the sharp increase in the use of Korean herbal medicine covered by auto insurance following traffic accidents, focusing on whether this trend reflects the opportunistic behavior of Korean herbal medicine doctors.

Methods

We extend the theoretical framework of physician-induced demand by incorporating differences in marginal costs between medical and herbal doctors. Using the Korea Health Panel data from 2017, we apply a recently developed decomposition method to compare utilization patterns between conventional and herbal medicine providers treating the same types of injuries.

Results

The analysis identifies significant structural differences in treatment behavior between herbal and medical doctors. At higher quantiles of medical utilization, herbal medicine shows disproportionately greater service provision, consistent with more substantial opportunistic incentives. Changes in the relationship between outpatient visits and structural differences account for 8.96 visit differences (133%) of 6.697 visits, a 9.302 visit increase (102%) of 9.064 visits, and 8.309 visit differences (115%) of 7.186 visits in the 7th, 8th, and 9th deciles, respectively. Changes in the relationship of hospital stay to the structural differences account for seven-day differences (442%) of a 1.5-day increase in the 7th decile, two days (300%) of a 1-day decrease in the 8th decile, and 8.842 days (186%) of a 4.75-day increase in the 9th decile. These results remain robust after controlling for patient characteristics and injury severity.

Conclusions

Findings suggest that herbal medicine services under auto insurance are more structurally driven by provider behavior than patient need, implying a need for closer scrutiny of reimbursement and review procedures to mitigate opportunistic practice.

---

## [Decision Letter · Decision Letter 1]

24 Feb 2026

PONE-D-25-53640R1Physician behavior for “invisible” treatment; Korean herbal medicine doctor's treatment covered by auto insurancePLOS One

Dear Dr. Lee,

Thank you for submitting your manuscript to PLOS ONE. After careful consideration, we feel that it has merit but does not fully meet PLOS ONE’s publication criteria as it currently stands. Therefore, we invite you to submit a revised version of the manuscript that addresses the points raised during the review process.

We look forward to receiving your revised manuscript.

Kind regards,

Pasyodun Koralage Buddhika Mahesh

Academic Editor

PLOS One

Journal Requirements:

Reviewers' comments:

Reviewer's Responses to Questions

**Comments to the Author**

1. If the authors have adequately addressed your comments raised in a previous round of review and you feel that this manuscript is now acceptable for publication, you may indicate that here to bypass the “Comments to the Author” section, enter your conflict of interest statement in the “Confidential to Editor” section, and submit your "Accept" recommendation.

Reviewer #1: (No Response)

Reviewer #2: (No Response)

2. Is the manuscript technically sound, and do the data support the conclusions?

Reviewer #1: Partly

Reviewer #2: Partly

3. Has the statistical analysis been performed appropriately and rigorously?

Reviewer #1: Yes

Reviewer #2: No

4. Have the authors made all data underlying the findings in their manuscript fully available?

Reviewer #1: Yes

Reviewer #2: No

5. Is the manuscript presented in an intelligible fashion and written in standard English?

Reviewer #1: No

Reviewer #2: Yes

6. Review Comments to the Author

Reviewer #1: Dear Dr. Lee,

Please note that my comments have been attached separately. I wish you all the best.

Thank you

Reviewer #2: This manuscript addresses an important and relevant topic concerning healthcare utilization and provider behaviour under auto insurance coverage. The use of nationally representative Korea Health Panel data enhances the external validity and generalizability of the findings. The study is supported by a strong theoretical foundation based on physician-induced demand and health economics principles. Applying a decomposition method to examine structural differences in utilisation patterns is methodologically appropriate and provides useful insight into potential differences between herbal and conventional medicine providers. The results are clearly presented using tables and figures, and the findings contribute to policy discussions regarding reimbursement systems and provider incentives.

However, several deficiencies need to be addressed:

• Absence of a separate Methods section: The manuscript does not include a clearly defined Methods section. Instead, elements of study design, empirical methods, and data description are dispersed across the Introduction, theoretical background, and separate sections such as “Decomposition” and “Data.” This structure reduces clarity and makes it difficult for readers to fully understand and replicate the study methodology.

• Study design not explicitly stated: The manuscript does not clearly identify the study as an observational cross-sectional secondary data analysis, which is essential for transparency and appropriate interpretation.

• Incomplete description of participants: The total sample size, inclusion and exclusion criteria, and the number of participants in each comparison group are not clearly reported, limiting assessment of representativeness and potential selection bias.

• Insufficient discussion of bias and confounding: Potential sources of bias, including residual confounding, selection bias, and limitations inherent to secondary data analysis, are not adequately addressed.

• Unclear variable definitions: The exposure and outcome variables are not explicitly defined in sufficient detail, which affects reproducibility and interpretability.

• Overinterpretation of findings: Some conclusions attribute structural differences directly to opportunistic behaviour without fully acknowledging the limitations of causal inference in observational studies.

• Limited discussion of study limitations: The limitations section does not comprehensively address key methodological constraints, including lack of clinical severity measures and possible unmeasured confounding.

• Inconsistent terminology: Multiple terms such as “normal doctors,” “usual doctors,” “ordinary doctors,” and “conventional doctors” are used interchangeably, which may create confusion and reduce clarity.

• Use of first-person narrative: The manuscript uses first-person expressions such as “I refer to” and “I present,” which are inconsistent with standard scientific writing conventions and reduce the objectivity of the manuscript.

7. PLOS authors have the option to publish the peer review history of their article (what does this mean?). If published, this will include your full peer review and any attached files.

Reviewer #1: No

Reviewer #2: **Yes:**I.O.K.K.Nanayakkara

---

## [Author Response · Author response to Decision Letter 2]

14 Apr 2026

[Response to Reviewer #1]

Major Comment 1: Justification for the Chernozhukov et al. (2013) method.

Response: We thank the reviewer for this insightful suggestion. We have revised the manuscript (Page 8) to clearly articulate the advantages of the Chernozhukov et al. (2013) method over the traditional Oaxaca-Blinder approach. Specifically, we now emphasize that while the traditional method only captures differences in the mean, the functional decomposition method allows us to analyze the entire distribution (quantiles). This is crucial for our hypothesis, as opportunistic provider behavior—represented here as a structural effect—is more likely to be prevalent in the higher deciles of healthcare utilization. We have also clarified the roles of endowment effects and structural effects to make the methodology more accessible to a broader audience.

Major Comment 2: Clarifying “Structural Effects.”

Response: We have added a brief explanation in the Methods section. We now clarify that Endowment Effects refer to differences in patient characteristics (e.g., age, education, income), while Structural Effects represent the 'unexplained' gap—which, in this study, we attribute to the differing incentive structures and behaviors of Korean Herbal Medicine (KHM) vs. Conventional Medicine (CM) providers.

Major Comment 3: Clarifying “Opportunistic behavior of the physician.”

Response: We now frame the structural differences as being indicative of supply-side incentives rather than definitive proof of opportunistic behavior. We have also explicitly acknowledged the limitations of causal inference in our observational study design.

Major Comment 4: Clarifying “Injury severity.”

Response: The variable named ‘disability’ is not a reflection of the severity of injury. We don’t have any information on the severity of injury in the data. We added an explanation of the ‘disability’ definition in the Variable section. We have also removed the injury severity from the manuscript to avoid confusion.

Major Comment 5: Regarding Table 1 and 2

Response: We explicitly define "LOS" (Length of Stay), "Income Quintiles," and rename "outpatient visit no" to "Number of Outpatient Visits." We added an explanation of the ‘disability’ definition in the Variable section. ‘Disability’ dummy indicates whether the respondent is classified as having a disability by the government.

Major Comment 6: Regarding the observation number of the data.

Response: We have added details on how many observations are used for this study in the data section. For the outpatient covered by auto insurance, 187 observations out of 301,540 are observed, and for the inpatient covered by auto insurance, 95 observations out of 3521 are observed in the 2017 KHP.

Major Comment 7: Certain conditions in the theoretical background.

Response: We have added a statement that our empirical results indicate the marginal utility of KHM doctors in inducing demand may exceed that of CM doctors, consistent with the theoretical background in the manuscript.

Major Comment 8 & 9: Reporting the ratio of conditional to observable outcomes.

Response: We have revised the reporting of our results on Pages 13 and 15 to enhance clarity. We now explicitly state that we are reporting the ratio of the structural (conditional) effect to the total observed difference. We have also rephrased the results to clearly reflect that positive structural values signify provider-side drivers of utilization, while negative endowment values indicate that patient characteristics do not explain the observed gap. This clarification ensures that the significance of these values is immediately accessible to the reader.

Minor Comments: Use of language, Labelling figures, tables and equations,

Response: We consistently use Conventional Medicine (CM) doctors and Korean Herbal Medicine (KHM) doctors. We also changed the labeling accordingly. We have switched entirely to an impersonal voice (e.g., "This study finds...") or to a consistent "We" if the journal allows it. We have converted all citations to Vancouver Style as required by PLOS ONE.

[Response to Reviewer #2]

Comment 1: Absence of a separate Methods section.

Response: We sincerely apologize for the lack of clarity. Following your suggestion, we have created a dedicated 'Methods' section which consolidates empirical methods, data and variables previously dispersed throughout the manuscript.

Comment 2: Study design not explicitly stated.

Response: We have now explicitly identified the study as a cross-sectional secondary data analysis in both the Abstract and the new Methods section.

Comment 3: Incomplete description of participants.

Response: We have added details on how many observations are used for this study in the data section. For the outpatient covered by auto insurance, 187 observations out of 301,540 are observed, and for the inpatient covered by auto insurance, 95 observations out of 3521 are observed in the 2017 KHP.

Comment 4: Insufficient discussion of bias and confounding.

Response: We have expanded the Limitations section to include a discussion on the lack of clinical severity measures and the potential for unmeasured confounding.

Comment 5: Unclear variable definitions.

Response: We have added a variable section and explained the variables used in the study.

Comment 6: Overinterpretation of findings.

Response: We have carefully revised the Results and Conclusions sections to soften our claims. We now frame the findings as "indicative of structural incentives" rather than "direct proof of opportunistic behavior," acknowledging the constraints of observational research.

Comment 7: Limited discussion of study limitations.

Response: We have expanded the Limitations section to include a discussion on the lack of clinical severity measures and the potential for unmeasured confounding.

Comment 8: Inconsistent terminology (Normal, Usual, etc.).

Response: We sincerely apologize for the confusion. We have standardized all terms to "Conventional Medicine (CM) doctors" and “Korean Herbal Medicine (KHM) doctors” throughout the revised manuscript.

Comment 9: Use of first-person narrative

Response: We have switched entirely to an impersonal voice (e.g., "This study finds...") or a consistent "We" if the journal allows it.

---

## [Decision Letter · Decision Letter 2]

15 May 2026

Physician behavior for “invisible” treatment; Korean herbal medicine doctor's treatment covered by auto insurance

PONE-D-25-53640R2

Dear Dr. Lee,

We’re pleased to inform you that your manuscript has been judged scientifically suitable for publication and will be formally accepted for publication once it meets all outstanding technical requirements.

Kind regards,

Pasyodun Koralage Buddhika Mahesh

Academic Editor

PLOS One

Additional Editor Comments (optional):

Reviewers' comments:

Reviewer's Responses to Questions

**Comments to the Author**

1. If the authors have adequately addressed your comments raised in a previous round of review and you feel that this manuscript is now acceptable for publication, you may indicate that here to bypass the “Comments to the Author” section, enter your conflict of interest statement in the “Confidential to Editor” section, and submit your "Accept" recommendation.

Reviewer #1: All comments have been addressed

Reviewer #2: All comments have been addressed

2. Is the manuscript technically sound, and do the data support the conclusions?

Reviewer #1: (No Response)

Reviewer #2: Yes

3. Has the statistical analysis been performed appropriately and rigorously?

Reviewer #1: (No Response)

Reviewer #2: Yes

4. Have the authors made all data underlying the findings in their manuscript fully available?

Reviewer #1: (No Response)

Reviewer #2: Yes

5. Is the manuscript presented in an intelligible fashion and written in standard English?

Reviewer #1: (No Response)

Reviewer #2: Yes

6. Review Comments to the Author

Reviewer #1: Dear Dr. Lee,

Thank you very much for your impressive work and for considering my suggestions in the revised version of your manuscript.

All the very best to you and your colleagues!

Reviewer #2: The authors have carefully revised the manuscript and addressed all reviewer comments appropriately. I have no additional suggestions and recommend acceptance.

7. PLOS authors have the option to publish the peer review history of their article (what does this mean?). If published, this will include your full peer review and any attached files.

Reviewer #1: **Yes:**Herath Mudiyanselage Chathurika Dulmini Herath

Reviewer #2: **Yes:**I.O.K.K.Nanayakkara

---

## [Editor Report · Acceptance letter]

PONE-D-25-53640R2

PLOS One

Dear Dr. Lee,

I'm pleased to inform you that your manuscript has been deemed suitable for publication in PLOS One. Congratulations! Your manuscript is now being handed over to our production team.

Kind regards,

on behalf of

Dr. Pasyodun Koralage Buddhika Mahesh

Academic Editor

PLOS One